# Prevention and Treatment of Venous Thromboembolism Associated with Amivantamab-Based Therapies in Patients with Lung Cancer—Provisional Clinical Opinion Based on Existing Clinical Practice Guidelines

**DOI:** 10.3390/cancers17020259

**Published:** 2025-01-14

**Authors:** Florian Moik, Gudrun Absenger, Robert Wurm, Maximilian J. Hochmair, Cihan Ay

**Affiliations:** 1Department of Internal Medicine, Division of Oncology, Medical University of Graz, 8036 Graz, Austria; florian.moik@medunigraz.at (F.M.); gudrun.absenger@medunigraz.at (G.A.); 2Department of Medicine I, Division of Haematology & Haemostaseology, Medical University of Vienna, Währinger Gürtel 18-20, 1090 Vienna, Austria; 3Department of Internal Medicine, Division of Pulmonology, Medical University of Graz, 8036 Graz, Austria; robert.wurm@uniklinikum.kages.at; 4Department of Respiratory and Critical Care Medicine, Karl Landsteiner Institute of Lung Research and Pulmonary Oncology, Klinik Floridsdorf, 1210 Vienna, Austria; maximilian.hochmair@gesundheitsverbund.at

**Keywords:** cancer, venous thromboembolism, thrombosis, amivantamab, lazertinib, lung cancer

## Abstract

Amivantamab-based therapies have recently been demonstrated to have superior efficacy compared to established treatment options in a specific subset of patients with lung cancer. In clinical trials evaluating amivantamab-based therapies, a high risk of venous thromboembolism was observed. Venous thromboembolism, including deep vein thrombosis and pulmonary embolism, is a known complication in patients with cancer and negatively affects the clinical course of disease. Increasing efforts are undertaken to predict and ultimately prevent cancer-associated venous thromboembolism. Here, we summarise available data on the risk of venous thromboembolism in patients treated with amivantamab-based therapies, and discuss this risk in the context of available guidelines on the prevention and treatment of venous thromboembolism in patients with cancer.

## 1. Introduction

Lung cancer is the leading cause of cancer-related death globally [1]. In recent years, the treatment options for patients with lung cancer have been dramatically improved by the implementation of subtype-specific targeted therapies and cancer immunotherapy. In patients with non-small cell lung cancer (NSCLC), activating alterations in the epidermal growth factor receptor (EGFR) are present in about 10% of patients, mostly those with adenocarcinoma [2]. Lately, different targeted therapies have been developed specifically for the treatment of patients with EGFR-altered NSCLC and have significantly improved the prognosis in this population [2].

Recently, amivantamab-based therapies have been investigated for the treatment of EGFR-mutated NSCLC. Amivantamab is a bispecific receptor-directed antibody targeting EGFR and mesenchymal–epithelial transition (MET) kinase. Increased efficacy was reported for amivantamab in combination with lazertinib, an EGFR-targeted tyrosine kinase inhibitor (TKI), compared to the established standard of care of osimertinib monotherapy in patients with untreated EGFR-mutated NSCLC in the MARIPOSA trial, and in combination with chemotherapy compared to chemotherapy alone after progression compared to osimertinib the MARIPOSA-2 trial [3,4]. Furthermore, enhanced efficacy has been reported in patients with EGFR exon 20 insertions for amivantamab and chemotherapy compared to chemotherapy alone in the first-line setting in the PAPILLON trial and for amivantamab compared to chemotherapy in the second-line setting [5,6]. Currently, amivantamab in combination with carboplatin and pemetrexed is approved for the first-line treatment of adult patients with advanced NSCLC with activating EGFR exon 20 insertion mutations and with EGFR common mutations, after the failure of prior therapy, including an EGFR TKI. Beyond this, a growing proportion of patients with NSCLC is anticipated to be treated with amivantamab-based therapies in the future.

Importantly, a very high risk of venous thromboembolism (VTE) was observed in clinical trials evaluating amivantamab-based therapies, which has led to the implementation of anticoagulation for primary thromboprophylaxis for the first 4 months of treatment in ongoing trials for amivantamab–lazertinib combination therapies [7]. The aim of this clinical opinion piece is to summarize available data on the risk of VTE in patients treated with amivantamab-based therapies and evaluate available clinical guidance on the use of primary pharmacological thromboprophylaxis with anticoagulation in ambulatory patients with cancer and treatment of established VTE in clinical practice.

## 2. Risk of VTE in NSCLC

Patients with cancer are at an increased risk of VTE compared to individuals without cancer [8]. The occurrence of VTE leads to interruptions and sometimes even discontinuation of anticancer therapies and increases morbidity and mortality [9,10]. Several patient-, cancer-, and treatment-related factors contribute to the risk of VTE [8,9]. The highest risk of cancer-associated VTE is observed among patients with advanced cancer stages and those undergoing systemic anticancer therapies [8,11]. Consistently, the risk of VTE in patients with NSCLC varies substantially according to the underlying cancer stage and type of systemic treatment [12]. In a Danish population-based study, the 1-year risk of VTE was 7.7% in patients with metastatic NSCLC initiating chemotherapy, 12.2% in those initiating immunotherapy, and 12.5% in those initiating targeted therapies [12]. In addition, the risk of VTE seems to vary according to different molecular subsets in NSCLC. Genomic alterations in anaplastic lymphoma kinase (ALK) and c-ros oncogene 1 (ROS1) have been previously identified to be associated with a very high risk of VTE, whereas lower rates were reported for EGFR-mutated and ALK/ROS1/EGFR-wildtype cancers [13,14,15,16,17]. Based on a meta-analysis of published cohorts, the overall incidences of VTE with ALK, ROS1, EGFR, and wildtype cancers are 41%, 30%, 12%, and 14%, respectively [14].

## 3. Risk of VTE with Amivantamab-Based Therapies

Recently published trials reported an increased risk of VTE in patients treated with amivantamab-based therapies compared to patients in the control arm. In detail, in the phase III MARIPOSA trial, including patients with untreated EGFR-mutated NSCLC, the risk of VTE at a median follow-up of 22 months was 37% in the amivantamab–lazertinib group (n = 429) and 9% in the osimertinib group (n = 429) [3]. In the amivantamab–lazertinib arm, the median time to VTE was 84 days (range: 6–777), with 62% of VTE events occurring within 4 months of treatment, 17% between months 5 and 8, and 21% beyond 9 months, implying an ongoing relevant risk of VTE in treated patients. Of 157 observed VTE events in the amivantamab–lazertinib arm, 110 were classified as grade 1–2 (70.1%), 47 were grade 3–5 (29.9%), 2 were fatal (1.3%), and 12 led to treatment discontinuation (7.6%) [3]. Further, in the phase III MARIPOSA-2 trial, patients with pre-treated EGFR-mutated NSCLC after disease progression to osimertinib were included [4]. After a median follow-up of 8.7 months, VTE was reported in 22% of patients receiving the combination of amivantamab–lazertinib and chemotherapy (n = 263), 10% in those with amivantamab and chemotherapy (n = 131), and 5% in those with chemotherapy alone (n = 263) [4,18]. The median time to VTE diagnosis in the amivantamab–lazertinib and chemotherapy arm was 63 days (range: 4–336) [4]. In the phase III PAPILLON trial, including patients with untreated NSCLC with EGFR exon 20 insertions, the reported rates of pulmonary embolism were 7.9% with amivantamab and chemotherapy and 4.5% with chemotherapy alone, whereas the rates of deep vein thrombosis were 6.6% and 1.9%, respectively [5,19]. Further, in a report of cohort C of the CHRYSALIS-2 trial, including patients with NSCLC and atypical EGFR mutations treated with amivantamab–lazertinib (n = 105), the reported risk of VTE was 30% [20]. In addition, in a joint analysis of single-arm trials, including 560 patients treated with amivantamab, 536 with amivantamab–lazertinib and 252 with lazertinib, the reported rate of VTE was 21% for amivantamab–lazertinib, 11% for amivantamab, and 11% for lazertinib [21]. In an exploratory analysis, risk factors for VTE were higher age, decreased performance status, and response to therapy [21]. Lastly, in the phase III PALOMA-3 trial, including patients with EGFR-mutated NSCLC after disease progression to osimertinib, a modest decrease in the risk of VTE in patients treated with subcutaneous amivantamab compared to intravenous amivantamab, both in combination with lazertinib, was observed [22]. In detail, the risks of VTE in patients receiving subcutaneous (n = 206) and intravenous treatment (n = 212) were 9% and 14%, respectively. However, based on earlier reports from other trials, primary prophylactic anticoagulation per local standards for the first 4 months of treatment was established during the conduct of the PALOMA-3 trial, which was used in 80% of patients in the subcutaneous treatment arm and 81% of those in the intravenous treatment arm. The most frequently used type of anticoagulation was direct factor Xa inhibitors in 64% and 68%, respectively. The risk of VTE in patients receiving prophylactic anticoagulation was 7% in the subcutaneous therapy group and 12% in the intravenous therapy group, compared to 17% and 26% in those not receiving anticoagulation, respectively. The risk of grade ≥3 bleeding with prophylactic anticoagulation was 2% in the subcutaneous and 0.6% in the intravenous treatment arm [22].

Synoptically, in patients treated with amivantamab–lazertinib combinations, the reported rates of VTE without prophylactic anticoagulation range between 21% and 37%, with considerable heterogeneity depending on follow-up duration and line of systemic treatment. In addition, reported rates of VTE during amivantamab-based therapies without lazertinib were 10–11%, with higher rates as opposed to the respective control-arm patients [4,5,21].

In Table 1, an overview of published data on the risk of VTE associated with amivantamab-based therapies is provided.

## 4. Primary Thromboprophylaxis in Ambulatory Patients with Cancer

Patients with cancer undergoing systemic antineoplastic therapies are at an increased risk of VTE [8,24]. However, due to the heterogeneity in risk between different subsets of patients and the concomitant underlying bleeding risk, unselected primary thromboprophylaxis is not recommended. In recent years, various predictive risk factors and biomarkers were identified and proposed for the identification of patients with cancer at high risk of VTE [24,25,26]. Subsequently, different risk assessment models and scores were developed to stratify VTE risk. Validated risk assessment models include the Khorana score, the Vienna CATS model, and the COMPASS-CAT score [25,27]. In Table 2, an overview of these models is provided. The Khorana score has been broadly validated, yet the performance of the Khorana score in patients with lung cancer has been demonstrated to be poor [28].

Primary thromboprophylaxis in ambulatory patients with cancer was consistently associated with a decreased risk of VTE and only with a modest increase in bleeding risk in the subset of patients at high baseline risk of VTE, indicating a net clinical benefit for these patients. In detail, in the phase III AVERT trial, patients with cancer at high risk for VTE (defined as Khorana score ≥ 2) were randomized to receive primary thromboprophylaxis with apixaban 2.5 mg twice daily or placebo for 6 months. At 6 months, VTE occurred in 4.2% of patients in the apixaban group and 10.2% in the placebo group, whereas major bleeding occurred in 3.5% and 1.8%, respectively [33]. Similarly, in the phase III CASSINI trial, patients with cancer initiating systemic therapies at high risk for VTE (defined as Khorana score ≥ 2) were randomized to receive rivaroxaban 10 mg daily or placebo for 6 months. During the intervention period, VTE occurred in 2.6% of patients in the rivaroxaban group and 6.4% in the placebo group, whereas major bleeding was observed in 2.0% and 1.0%, respectively [34]. In a meta-analysis of randomized controlled trials investigating primary thrombophylactic anticoagulation in ambulatory patients with cancer at high risk for VTE (defined as Khorana score ≥ 2), primary thromboprophylaxis was associated with around a 50% decrease in VTE risk both for prophylactic dose direct anti-Xa inhibitors and low-molecular-weight heparin (LMWH). Further, the risk of major bleeding was similar between groups in trials assessing LMWH, whereas a 2-fold increased risk was observed for patients receiving direct anti-Xa inhibitors, with an overall low absolute risk [35].

These developments have led to recommendations of risk-stratified primary thromboprophylaxis strategies in current guidelines on the management of VTE in patients with cancer [27,36,37]. Specifically, according to the current guidelines by the European Society for Medical Oncology (ESMO), patients with cancer should be offered an individualized risk assessment for VTE [27]. In patients with an estimated risk of VTE of >8–10% at 6 months, a discussion of primary thromboprophylaxis is suggested [27]. In ambulatory patients initiating systemic antineoplastic therapies at high risk for VTE, primary thromboprophylaxis with either a direct oral anti-Xa inhibitor or LMWH for a maximum of 6 months may be considered [27]. In Table 3, an overview of anticoagulants evaluated in the setting of primary thromboprophylaxis in patients with cancer and listed within the ESMO guidelines is provided.

## 5. Proposed Clinical Approach for the Prevention of VTE During Amivantamab-Based Therapies

Following the available guideline recommendations for the prevention and management of cancer-associated VTE, an individualized risk assessment for the concomitant risks of VTE and bleeding should be conducted prior to the initiation of amivantamab-based therapies in patients with NSCLC. Based on the consistent reports of very high rates of VTE associated with amivantamab–lazertinib combinations, especially early during treatment, which exceed the 6-month threshold of 8–10% stated in the ESMO guidelines, primary thromboprophylaxis should be considered in the absence of contraindications [27]. This approach is supported by data from the PALOMA-3 trial, which demonstrated a lower risk of VTE in patients receiving primary thromboprophylaxis [22]. Further, a reassuringly low risk of bleeding events was observed in patients receiving prophylactic anticoagulation in PALOMA-3 [22]. Moreover, a modest decrease in the risk of VTE was observed in the subcutaneous as compared to the intravenous treatment arm in PALOMA-3. However, the risk of VTE in patients not receiving prophylactic anticoagulation in the subcutaneous treatment arm was still high and reached 17% over a median duration of treatment of 4.7 months [22]. Therefore, primary thromboprophylaxis in patients treated with amivantamab–lazertinib for the first 6 months of therapy seems to be justified in accordance with ESMO and other international guidelines, irrespective of the mode of treatment administration [27,36,37]. Beyond the initial 6-month treatment timeframe, no dedicated data or guidance for primary thromboprophylaxis strategies is currently available.

Importantly, primary prophylactic anticoagulation should be carefully assessed, considering the competing risks of VTE and bleeding. Therefore, in patients with present bleeding risk factors, primary thromboprophylaxis must be used with caution. Known risk factors for bleeding in patients with cancer include severe coagulation or platelet function abnormalities, liver or kidney function abnormalities, a recent history of severe bleeding, higher age, a higher risk for falls, and severe thrombocytopenia [26,38]. In addition, the use of direct anti-Xa inhibitors was associated with a higher risk of gastrointestinal and genitourinary bleeding as opposed to LMWH in the treatment of cancer-associated VTE [39]. Therefore, the use of direct oral anti-Xa inhibitors should be critically evaluated in the case of elevated bleeding risk due to gastrointestinal mucosal disturbances or an elevated risk of genitourinary bleeding. Accordingly, in the ESMO guidelines, the use of LMWH in the case of safety concerns of oral direct anti-Xa inhibitors and a clinically relevant risk of VTE is suggested [27].

Additional challenging clinical scenarios and predisposing bleeding risk factors specifically relevant to the population of patients with lung cancers should be considered. First, concomitant use of antiplatelet therapy is frequent in patients with lung cancer based on the high co-prevalence of cardiovascular diseases [40]. Therefore, antiplatelet therapy should be considered as a potential additional bleeding risk factor when evaluating the risk–benefit ratio of primary prophylactic anticoagulation. Secondly, patterns of bleeding events specific to patients with lung cancer might further impact the application of prophylactic anticoagulation. Hemoptysis is common in patients with lung cancer [41,42,43]. No specific data are available on the impact of prophylactic anticoagulation in patients with prior hemoptysis, yet a recent severe bleeding event is considered a relative contraindication to anticoagulation [44]. Further, brain metastases are common among patients with EGFR-mutated NSCLC, with up to 60% of patients diagnosed over the course of the disease [45]. Brain metastases are associated with intracranial hemorrhage, yet a lower risk of bleeding was reported for patients with lung cancer compared to other cancer types [46]. However, the impact of anticoagulation on the risk of intracranial bleeding in patients with brain metastasis is currently unclear. Observational data, including of patients with brain metastasis, suggest no increase in the risk of intracranial hemorrhage in patients receiving therapeutic LMWH and the comparable safety of therapeutic dose LMWH and direct factor Xa inhibitors [46,47]. In the absence of dedicated studies, general suggestions and contraindications for selecting patients for anticoagulation should, therefore, be followed.

In patients treated with amivantamab-based therapies without lazertinib, unclarity exists regarding the clinical consequence of the observed VTE risk in the context of available guidance. In the published phase III trials, a higher risk of VTE was observed in the amivantamab-arm as opposed to control-arm patients, yet the risk of VTE at 6 months cannot be extrapolated from available data, and, therefore, no clear comparison to the proposed 8–10% 6-month risk threshold for considering primary thromboprophylaxis in the ESMO guidelines can be made. Therefore, in patients initiating amivantamab-based therapies without lazertinib, an individualized assessment of VTE risk with validated risk prediction models (e.g., Khorana score, Vienna CATS model) [27,31] to select patients for primary thromboprophylaxis may be considered according to the available guidance [27].

### Clinical Opinion

Patients with lung cancer initiating amivantamab-based therapies should undergo an individualized assessment of VTE risk, using a validated risk assessment model.Available guidelines suggest a risk-stratified approach to select ambulatory patients with cancer initiating systemic therapies for primary thromboprophylaxis. Based on the consistently high risk of VTE with amivantamab–lazertinib combinations, primary pharmacological thromboprophylaxis with a direct oral factor Xa inhibitor (apixaban and rivaroxaban) or LMWH for the first 6 months of therapy can be considered in the absence of an elevated risk of bleeding.Currently, unclarity exists regarding the timeframe beyond 6 months and the management of patients treated with amivantamab without lazertinib. In the latter setting, primary thromboprophylaxis may be considered in high-risk ambulatory cancer patients using validated risk models.Challenging clinical scenarios that might affect the risk–benefit ratio of primary thromboprophylaxis in patients with NSCLC include prior antiplatelet therapy, prior bleeding events, including hemoptysis, and the presence of brain metastasis. In the absence of dedicated studies, general principles of selecting patients for prophylactic anticoagulation should be followed.

## 6. Treatment of VTE During Amivantamab-Based Therapies

In patients diagnosed with VTE during the course of amivantamab-based therapies, treatment strategies should comply with the latest recommendations from available guidelines on anticoagulation therapy for VTE treatment in patients with cancer [27,36,37,48,49]. Briefly, therapeutic dose anticoagulation for at least 6 months should be used, considering extended anticoagulation for ongoing active malignancy and systemic anticancer treatment [27,36,37,50]. In Table 4, an overview of dosing regimens for anticoagulation therapy of cancer-associated VTE is provided.

Direct oral factor Xa inhibitors are associated with a decreased risk of VTE recurrence compared to LMWH in the treatment of cancer-associated VTE [39]. However, bleeding risk was higher for direct factor Xa inhibitors, which was mainly driven by gastrointestinal and genitourinary bleeding events. Therefore, if an elevated risk of gastrointestinal- or genitourinary bleeding is suspected, LMWH should be the preferred treatment option [27,36,37]. Further, amivantamab-based therapies might be resumed as soon as clinically feasible after initiation of anticoagulation therapy, yet no dedicated guidance or data exist to select the ideal timeframe for therapy resumption.

Importantly, a previous history of cancer-associated VTE was consistently reported as an independent and strong risk factor for subsequent VTE events in patients with cancer [24,51]. Anticoagulation treatment with therapeutic dose anticoagulation is recommended in patients with cancer-associated VTE for at least 6 months, with prolonged anticoagulation in the case of ongoing active malignancy or continuous anticancer therapies [27,36,37,48,49,52]. Therefore, ongoing anticoagulation therapy should be considered in patients with a history of prior cancer-associated VTE initiating amivantamab-based therapies.

### Clinical Opinion

In patients with cancer-associated VTE, a therapeutic dose of LMWH, apixaban, rivaroxaban, or edoxaban (following a lead-in phase of LMWH for at least 5 days) can be used for treatment in the acute phase. Unfractionated heparin (UFH) may be considered in patients with severe renal impairment. LMWH and the direct oral factor Xa inhibitors are preferred over vitamin K antagonists (VKAs) and should be continued for 6 months.Extended anticoagulation with LMWH, apixaban, rivaroxaban, or edoxaban beyond 6 months should be considered for patients with active cancer and/or those with continued systemic anticancer therapies if the expected bleeding risk is not higher than the risk of VTE recurrence. The decision to reduce to a prophylactic dose of the apixaban or rivaroxaban needs to be evaluated on a case-by-case basis.

## 7. Discussion

Amivantamab-based therapies are associated with increased efficacy compared to established treatment options in patients with EGFR-mutated NSCLC, yet a higher risk of VTE was observed in clinical trials. Importantly, the highest rates of VTE occurred during amivantamab–lazertinib combination therapies, and most VTE events occurred in the early phase of the therapy. However, available data suggest that the risk of VTE persists beyond the initial treatment timeframe.

Patients with advanced NSCLC have a known increased risk of VTE, especially in those receiving systemic anticancer therapies and in certain molecular subsets, including ALK- or ROS1-altered tumors [12,14]. Previously, similar rates of VTE were reported with EGFR compared to wildtype cancers. Therefore, the high risk of VTE observed with amivantamab-based therapy and the increased rates compared to control-arm patients seem to indicate a potential causal prothrombotic treatment-related effect. The underlying mechanisms contributing to the observed VTE risk with amivantamab-based therapies still remain to be elucidated.

Based on the early reports of high VTE rates in patients treated with amivantamab–lazertinib combinations, anticoagulation for primary thromboprophylaxis per physician choice was implemented in clinical trials [7]. Further, existing guidelines on primary thromboprophylaxis in ambulatory patients with cancer suggest primary thromboprophylaxis for the first 6 months after initiation of systemic treatment in patients with an expected VTE risk at 6 months of >8–10% [27]. This guidance is based on the risk–benefit calculation of the competing risks of VTE and bleeding in patients with cancer. Overall, primary thromboprophylaxis is associated with a 50% reduction in VTE risk in patients with a high baseline risk of VTE [35]. In parallel, a modestly increased risk of bleeding events in patients receiving prophylactic anticoagulation was observed, yet the absolute risk of bleeding was reportedly low. For example, 6-month risks of major bleeding of 2–3% were observed in large-scale thromboprophylaxis trials using anti-factor Xa inhibitors in patients at high risk of cancer-associated VTE [33,34]. The safety of primary thromboprophylaxis strategies is further supported by recent data from the PALOMA-3 trial, which found a low risk of bleeding in patients treated with amivantamab-based therapies who received prophylactic anticoagulation [22].

Several open questions regarding the ideal thromboprophylaxis strategy in patients treated with amivantamab-based therapies still need to be answered. First, the reported rates of VTE from phase III trials evaluating amivantamab-based therapies were stratified based on arbitrary timepoints (e.g., within 4 months after treatment initiation), and, currently, no data are available providing a dedicated time-to-event analysis of VTE risk in treated patients. This aspect limits the ability to adequately calculate the risk–benefit ratio or prophylactic anticoagulation for long-term prevention of VTE beyond the initial treatment phase of 6 months. Also, only limited data are available on the risk of VTE associated with amivantamab therapy without lazertinib, yet the VTE rates in published trial data suggest a modest increase as opposed to control-arm patients. However, these data do not currently allow the application of existing guideline recommendations for primary thromboprophylaxis in this setting. Thirdly, currently available data on VTE risk associated with amivantamab-based therapies is based on clinical trial data only, with no data available on the risk of VTE and/or bleeding in clinical practice cohorts, which might represent a generally more vulnerable population. Lastly, mechanistic data on the risk of VTE associated with amivantamab-based therapies are not available. Therefore, increasing efforts should be undertaken to explore the risk of VTE associated with amivantamab-based therapies in more granularity to further improve the ability to predict and ultimately prevent VTE in treated patients.

## 8. Conclusions

In conclusion, patients treated with amivantamab-based therapies have a high risk for VTE. Therefore, existing guidance on the use of primary thromboprophylaxis in ambulatory patients with cancer should be applied to help mitigate the risk of VTE.

## Figures and Tables

**Table 1 cancers-17-00259-t001:** Overview of data from published clinical trials reporting risk of VTE associated with amivantamab-based therapies.

Trial	Study Population	Median Follow-Up	Treatment (n)	Median Duration of Treatment	VTE	VTE Specifics	Prophylactic Anticoagulation
MARIPOSA phase III [3]	1st line— EGFR-mutated (exon 19 deletions or L858R mutation) N = 1074	22.0 months	Amivantamab + lazertinib (n = 429)	18.5 months (range 0.2–31.4)	37% (157/421)	62% within 4 months	At baseline: 5%; at the time of VTE: 1%
			Osimertinib (n = 429)	18.0 months (range 0.2–32.7)	9% (39/428)	33% within 4 months	At baseline: 5%; at the time of VTE: 0%
			Lazertinib (n = 216)	n.r.	n.r.	n/a	-
PAPILLON phase III [5]	1st line— EGFR exon 20 insertions N = 308	14.9 months	Amivantamab + chemotherapy (n = 153)	9.7 months (range: 0.1–26.9)	n.r.	PE: 7.9% DVT: 6.6%	n.r.
			Chemotherapy (n = 155)	6.7 months (range: 0–25.3)	n.r.	PE: 4.5% DVT: 1.9%	n.r.
MARIPOSA-2 (phase III) [4]	EGFR-mutated (exon 19 deletions or L858R) PD after osimertinib N = 657	8.7 months	Amivantamab + lazertinib + chemotherapy (n = 263)	5.8 months (range 0.1–18.6 months)	22% (58/263)	Median time to VTE: 63 days (range: 4–336)	At the time of VTE: 2%
			Amivantamab + chemotherapy (n = 131)	6.3 months (range 0–14.7 months)	10% (13/130)	Median time to VTE: 71 days (range: 15–233)	At the time of VTE: 0%
			Chemotherapy (n = 263)	3.7 months (range 0–15.9 months)	5% (11/243)	Median time to VTE: 43 days (range: 9–92)	At the time of VTE: 1%
PALOMA-3 phase III [22]	EGFR-mutated, PD after osimertinib and chemotherapy N = 418	7.0 months (range: 0.1–14.4)	Amivantamab iv + lazertinib (n = 212)	4.1 months (range, 0.0–13.2)	14%	67% within 4 months	80%; VTE with vs without prophylactic anticoagulation: 12% vs. 26%
			Amivantamab sc + lazertinib (n = 206)	4.7 months (range, 0.1–13.2)	9%	74% within 4 months	81%; VTE with vs without prophylactic anticoagulation: 7% vs. 17%
Post hoc analysis of 3 trials (CHRYSALIS, CHRYSALIS-2, and LASER201) (meeting abstract) [21]	Various, predominantly in the TKI-relapsed setting N = 560	n.r.	Amivantamab + lazertinib (n = 536)	n.r.	21%	Median time to VTE: 79 days; 64% within 4 months	n.r.
			Amivantamab (n = 560)	n.r.	11%	Median time to VTE: 85 days	n.r.
			Lazertinib (n = 252)	n.r.	11%	Median time to VTE: 170 days	n.r.
CHRYSALIS-2 cohort C phase I/Ib (meeting abstract) [20]	Atypical EGFR mutations, excluding exon 20 insertions; treatment-naïve or had ≤2 prior lines, which may have included a 1st/2nd-generation EGFR TKI N = 105	13.8 months (range: 0.1–30.2)	Amivantamab + lazertinib (n = 105)	n.r.	30% (31/105)	71% within 4 months	At the time of VTE: 3%
CHRYSALIS phase I [23]	EGFR mutant NSCLC after PD to 3rd gen. TKI	13.3 months (range: 0.5–23.7)	Amivantamab + lazertinib (n = 91; safety population)	n.r.	n.r.	PE: 12%	n.r.

**Table legend**: abbreviations: DVT: deep vein thrombosis; EGFR: epidermal growth factor receptor; iv: intravenous; n/a: not applicable; n.r.: not reported; NSCLC: non-small cell lung cancer; PE: pulmonary embolism; PD: progressive disease; sc: subcutaneous; TKI: tyrosine kinase inhibitor; VTE: venous thromboembolism.

**Table 2 cancers-17-00259-t002:** Selected validated risk assessment models for cancer-associated VTE.

Model	Items	Cut-Off for Defining High VTE Risk
Khorana score [29,30]	Cancer type category (very high risk: +2 points, high risk: +1 point, low risk: +0 points) * Hb < 10 g/dl or ESA use: +1 point WBC >11 G/L: + 1 point Platelets ≥350 G/L: +1 point BMI >35 kg/m^2^: +1 point	Original cut-off: ≥3 points; validation and interventional studies: ≥2 points
Vienna CATS model [31]	Nomogram-based individualized prediction of 6-month VTE risk based on cancer type category and continuous D-Dimer levels **	Predicted 6-month risk of VTE ≥ 10%
COMPASS-CAT [32]	Anthracycline treatment: +6 points Time since cancer diagnosis ≤6 months: +4 points Central venous catheter use: 3 points Advanced stage of cancer: 2 points Cardiovascular risk factors present ***: +5 points Recent hospitalization for acute medical illness: +2 points History of VTE: +1 point Platelet count ≥350 G/L: +2 points	≥7 points

**Table legend**: abbreviations: BMI: body mass index; ESA: erythropoiesis-stimulating agents; Hb: hemoglobin; VTE: venous thromboembolism; WBC: white blood cell count. * Very-high-risk cancer types, according to the Khorana score, include pancreatic and gastric cancer; high-risk cancer types include lung, lymphoma, gynecologic, bladder, and testicular cancer; and low-risk cancer types include breast and prostate cancer. ** Individual 6-month risk of VTE calculated via a nomogram [31]. *** Cardiovascular risk factors in the COMPASS-CAT score include two or more of the following: history of peripheral artery disease, ischemic stroke, coronary artery disease, hypertension, hyperlipidemia, diabetes, and obesity.

**Table 3 cancers-17-00259-t003:** Options for pharmacological thromboprophylaxis in ambulatory patients with cancer.

Agent	Dosing
**Direct anti-Xa inhibitors**	
Apixaban	2.5 mg twice daily orally
Rivaroxaban	10 mg once daily orally
**Low-molecular-weight heparin**	
Bemiparin	3500 anti-Xa IU once daily s.c.
Dalteparin	5000 anti-Xa IU once daily s.c.
Enoxaparin	4000 anti-Xa IU once daily s.c.
Nadroparin	3800 anti-Xa IU (if weight > 70 kg: 5700 anti-Xa IU) once daily s.c.
Tinzaparin	4500 anti-Xa IU once daily s.c.
**Selective parenteral indirect factor Xa inhibitor**	
Fondaparinux	Not studied in the outpatient prophylaxis setting

**Table legend**: adapted from ESMO clinical practice guidelines on VTE in cancer [27]. Listed agents have been evaluated in the setting of primary thromboprophylaxis in patients with cancer yet lack a specific indication for cancer outpatients in the package inserts. Abbreviations: IU: international units; kg: kilogram bodyweight; s.c.: subcutaneous.

**Table 4 cancers-17-00259-t004:** Options for therapeutic anticoagulation in patients with cancer-associated VTE.

Direct Anti-Xa Inhibitors	
Apixaban	10 mg twice daily for 7 days, followed by 5 mg twice daily orally
Rivaroxaban	15 mg twice daily for 3 weeks, followed by 20 mg once daily orally
Edoxaban	Lead-in period of at least 5 days of therapeutic LMWH, followed by 60 mg once daily orally (30 mg once daily orally if dose reduction criteria apply *)
**Low-molecular-weight heparin**	
Dalteparin	100 anti-Xa IU/kg twice daily, or 200 anti-Xa IU/kg once daily for the first 30 days, followed by 150 anti-Xa IU/kg once daily after day 30 s.c.
Enoxaparin	100 anti-Xa IU/kg twice daily, or 150 anti-Xa IU/kg once daily s.c.
Tinzaparin	175 anti-Xa IU/kg once daily s.c.

**Table legend**: adapted from ESMO clinical practice guidelines on VTE in cancer [27]. Abbreviations: CrCl, creatinine clearance; IU: international units; kg: kilogram bodyweight; LMWH: low-molecular-weight heparin; s.c.: subcutaneously. * Dose reduction criteria include CrCl < 50 mL/min, weight ≤ 60 kg, or patients receiving P-glycoprotein inhibitors.

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
