# Peer review of "Prevention and Treatment of Venous Thromboembolism Associated with Amivantamab-Based Therapies in Patients with Lung Cancer—Provisional Clinical Opinion Based on Existing Clinical Practice Guidelines"

_cancers, 2025, doi:10.3390/cancers17020259_

Round 1

Reviewer 1 Report

Comments and Suggestions for Authors

This manuscript tackles a clinically important issue: the prevention and treatment of venous thromboembolism (VTE) in patients receiving amivantamab-based therapies for lung cancer. The authors are to be commended for their time and effort in synthesizing evidence and providing practical guidance on this emerging topic. The manuscript offers valuable insights and is well-supported by existing clinical data and guidelines, making it a significant contribution to the field.

However, a few minor issues need to be addressed to enhance clarity and ensure consistency:

  1. Tables 3 and 4 may be unnecessary, as the dosing regimens for anticoagulation are well-documented in ESMO guidelines. If you believe these tables enhance the manuscript's value, retaining them is acceptable.
  2. Correct the typo in line 365, where "tables" is used incorrectly.
  3. Revise line 204-205 to read: "However, the risk of VTE in patients."
  4. Update line 213-214 to: "Therefore, in patients with present bleeding risk factors." Clarify whether this refers to one or multiple factors.
  5. Adjust line 228-229 to: "Antiplatelet therapy should be considered as a potential additional bleeding risk factor."
  6. Ensure consistent use of "metastasis" (singular or plural) in lines 235, 237, 240, and 270, and adjust verbs accordingly.
  7. Fix the singular-plural mismatch in line 245-246: "unclarity exists."
  8. Replace "ESMIO guidelines" in line 251 with "ESMO guidelines."
  9. Revise line 64 to: "of established VTE."
  10. Simplify line 68 by removing the unnecessary comma: "anti-cancer therapies and increases morbidity."
  11. Remove the comma in line 310: "systemic anti-cancer therapies if the expected bleeding risk."
  12. Update line 109 to: "Lastly, in phase III of the PALOMA-3 trial."
  13. Adjust line 137 to: "occurred at 2.6% in the rivaroxaban group."
  14. Revise line 173 to: "Further, the risk of major bleeding was similar."
  15. Edit line 179 to: "should be offered an individualized risk."
  16. Update line 239 to: "anticoagulation on the risk of intracranial."
  17. Correct line 275 to: "comply with the latest recommendations."
  18. Change line 75 to: "In addition, the risk of VTE."

By addressing these minor revisions, the manuscript will become more polished and accessible to readers. Thank you again for your valuable contribution to this important area of oncology care.

Author Response

Reviewer: This manuscript tackles a clinically important issue: the prevention and treatment of venous thromboembolism (VTE) in patients receiving amivantamab-based therapies for lung cancer. The authors are to be commended for their time and effort in synthesizing evidence and providing practical guidance on this emerging topic. The manuscript offers valuable insights and is well-supported by existing clinical data and guidelines, making it a significant contribution to the field. However, a few minor issues need to be addressed to enhance clarity and ensure consistency.

  • Author response: We thank the reviewer for evaluating our manuscript and are glad the reviewer agrees with the clinical relevance and the general presentation of our paper. We also thank the reviewer for providing the raised minor issues which we have carefully considered and revised our manuscript accordingly.

Reviewer: Tables 3 and 4 may be unnecessary, as the dosing regimens for anticoagulation are well-documented in ESMO guidelines. If you believe these tables enhance the manuscript's value, retaining them is acceptable.

  • Author response: Thank you for this comment. We agree that dosing specifics of prophylactic and therapeutic anticoagulation in patients with cancer are readily available. We implemented these Tables based on the feedback from our lung cancer specialists, who pointed out that providing this information at a low threshold within the paper itself can help the correct clinical implementation of available guideline-conform management strategies and have therefore decided to include the dosing specifics for those less familiar. We hope this is acceptable for the reviewer.

Reviewer:

  • Correct the typo in line 365, where "tables" is used incorrectly.
  • Revise line 204-205 to read: "However, the risk of VTE in patients."
  • Update line 213-214 to: "Therefore, in patients with present bleeding risk factors." Clarify whether this refers to one or multiple factors.
  • Adjust line 228-229 to: "Antiplatelet therapy should be considered as a potential additional bleeding risk factor."
  • Ensure consistent use of "metastasis" (singular or plural) in lines 235, 237, 240, and 270, and adjust verbs accordingly.
  • Fix the singular-plural mismatch in line 245-246: "unclarity exists."
  • Replace "ESMIO guidelines" in line 251 with "ESMO guidelines."
  • Revise line 64 to: "of established VTE."
  • Simplify line 68 by removing the unnecessary comma: "anti-cancer therapies and increases morbidity."
  • Remove the comma in line 310: "systemic anti-cancer therapies if the expected bleeding risk."
  • Adjust line 167 to: "occurred at 2.6% in the rivaroxaban group."
  • Revise line 173 to: "Further, the risk of major bleeding was similar."
  • Edit line 179 to: "should be offered an individualized risk."
  • Update line 239 to: "anticoagulation on the risk of intracranial."
  • Correct line 275 to: "comply with the latest recommendations."
  • Change line 75 to: "In addition, the risk of VTE."

Author response: We thank the reviewer for raising these issues, we have implemented all suggested changes in the revised manuscript.

Reviewer: Update line 109 to: "Lastly, in phase III of the PALOMA-3 trial."

Author response: The notation of “phase III trial” refers to the trial design in general of the PALOMA-3 trial, therefore this has not been changed.

Reviewer: By addressing these minor revisions, the manuscript will become more polished and accessible to readers. Thank you again for your valuable contribution to this important area of oncology care.

Author response: Thank you for your reviewer and for thereby helping us improve our manuscript.

Reviewer 2 Report

Comments and Suggestions for Authors

Cancer is known to cause deep vein thrombosis (DVT).  It has been noted that the recently developed anti-tumor treatment using amivantamab, although effective, has various side-effects including exacerbating DVT in non-small cell lung cancer (NSCLC) patients.  Based on the available data, these authors have provided an interim guideline for treating NSCLC cancer patients with venous thrombosis as well as for prophylactic treatments of the lung cancer patients using various anti-thrombotic agents.  The proposed treatment recommendations are not based on scientific mechanism-oriented research results but are based on the data from various clinical trials.  Overall, the recommendations proposed are reasonable and can be a good starting point.

In the Title, what is “an Austrian Interdisciplinary Focus Group”?  If it is an officially established group, please provide a brief description of the focus group.  For example, is it a subcommittee of a professional organization, publicly funded study group, etc.?  If it is a private ad hoc group, please consider deleting “from an Austrian Interdisciplinary Focus Group” from the title.

Please define all the acronyms used in the tables.

The document as a whole is a bit wordy.  Please try to shorten the text.  Some parts of Discussion is rather repetitious (i.e. repeating points made in previous sections).  

Author Response

Reviewer: Cancer is known to cause deep vein thrombosis (DVT).  It has been noted that the recently developed anti-tumor treatment using amivantamab, although effective, has various side-effects including exacerbating DVT in non-small cell lung cancer (NSCLC) patients.  Based on the available data, these authors have provided an interim guideline for treating NSCLC cancer patients with venous thrombosis as well as for prophylactic treatments of the lung cancer patients using various anti-thrombotic agents.  The proposed treatment recommendations are not based on scientific mechanism-oriented research results but are based on the data from various clinical trials.  Overall, the recommendations proposed are reasonable and can be a good starting point.

Author response: We thank the reviewer for evaluating our manuscript. We are glad the reviewer agrees with the importance of our work and the general conceptualisation of our proposed management strategies based on available guidelines.

Reviewer: In the Title, what is “an Austrian Interdisciplinary Focus Group”?  If it is an officially established group, please provide a brief description of the focus group.  For example, is it a subcommittee of a professional organization, publicly funded study group, etc.?  If it is a private ad hoc group, please consider deleting “from an Austrian Interdisciplinary Focus Group” from the title.

Author response: We used this unofficial label to describe our interdisciplinary collaborative effort in this project, which is based on an ongoing collaboration of multiple clinicians and researchers in the field of lung cancer in Austria. This collaboration is not related to any officially established group or organisation. We agree with the reviewers´ suggestion to change the title and have therefore revised the title as follows:

Original: Prevention and Treatment of Venous Thromboembolism Associated with Amivantamab-Based Therapies in Patients with Lung Cancer – Provisional Clinical Opinion from an Austrian Interdisciplinary Focus Group

Revision: Prevention and Treatment of Venous Thromboembolism Associated with Amivantamab-Based Therapies in Patients with Lung Cancer – Provisional Clinical Opinion based on existing Clinical Practice Guidelines

We believe this title more accurately reflects the nature of our work and hope the reviewer agrees with these changes.

Reviewer: Please define all the acronyms used in the tables.

Author response: All acronyms are now defined in the Table legends.

Reviewer: The document as a whole is a bit wordy.  Please try to shorten the text.  Some parts of Discussion is rather repetitious (i.e. repeating points made in previous sections). 

Author response: Thank you for your evaluation of the manuscript length. We have carefully reconsidered the composition of the discussion section (i.e., ca. 1 page), which summarizes the key elements of the manuscript and discussed some key limitations of our current state of knowledge. We therefore believe the discussion represents an important overview and were not able to significantly shorten this part without losing key information, vital to the presentation of the paper. We hope the reviewer agrees with these considerations and again thank the reviewer for evaluating our manuscript.